Mammalian lures monitored with time-lapse cameras increase detection of pythons and other snakes

McCampbell Marina 1
Spencer McKayla 2
Hart Kristen 3
Link Gabrielle 1
Watson Andrew 1
McCleery Robert 1 ramccleery@ufl.edu
1 Department of Wildlife Ecology and Conservation, University of Florida , Gainesville, Florida , United States of America
2 Florida Fish and Wildlife Conservation Commission , Gainesville, Florida , United States of America
3 Wetland and Aquatic Research Center, United States Geological Survey , Fort Lauderdale, Florida , United States of America
Baxter-Gilbert James
Electronic publication date: 2024 Jun 24
Publication date: 2024
Volume: 12
Electronic Location ID: e17577
Received 2023 Sep 21; Accepted 2024 May 24
Copyright year: 2024
License: This is an open access article, free of all copyright, made available under the Creative Commons Public Domain Dedication. This work may be freely reproduced, distributed, transmitted, modified, built upon, or otherwise used by anyone for any lawful purpose.
License URL: https://creativecommons.org/publicdomain/zero/1.0/

Keywords: Invasion, Detection dog, Everglades, Rabbit, Live lure

Funding: Florida Fish and Wildlife Conservation Commission #13416-A3044 United States Geological Survey #G19AC00432 Funding for this project was provided by the Florida Fish and Wildlife Conservation Commission (grant #13416-A3044) as well as the United States Geological Survey (grant #G19AC00432). The funders had no role in study design, data collection and analysis, decision to publish, or preparation of the manuscript.

==============================
Background

Enhancing detection of cryptic snakes is critical for the development of conservation and management strategies; yet, finding methods that provide adequate detection remains challenging. Issues with detecting snakes can be particularly problematic for some species, like the invasive Burmese python (Python bivittatus) in the Florida Everglades.

Methods

Using multiple survey methods, we predicted that our ability to detect pythons, larger snakes and all other snakes would be enhanced with the use of live mammalian lures (domesticated rabbits; Oryctolagus cuniculus). Specifically, we used visual surveys, python detection dogs, and time-lapse game cameras to determine if domesticated rabbits were an effective lure.

Results

Time-lapse game cameras detected almost 40 times more snakes (n = 375, treatment = 245, control = 130) than visual surveys (n = 10). We recorded 21 independent detections of pythons at treatment pens (with lures) and one detection at a control pen (without lures). In addition, we found larger snakes, and all other snakes were 165% and 74% more likely to be detected at treatment pens compared to control pens, respectively. Time-lapse cameras detected almost 40 times more snakes than visual surveys; we did not detect any pythons with python detection dogs.

Conclusions

Our study presents compelling evidence that the detection of snakes is improved by coupling live mammalian lures with time-lapse game cameras. Although the identification of smaller snake species was limited, this was due to pixel resolution, which could be improved by changing the camera focal length. For larger snakes with individually distinctive patterns, this method could potentially be used to identify unique individuals and thus allow researchers to estimate population dynamics.

Introduction

Snakes are reclusive and cryptic, often using areas that are hard for humans to access (e.g., fossorial, arboreal, and aquatic; Turner, 1977; Parker & Plummer, 1987; Durso, Willson & Winne, 2011). These characteristics make them difficult to detect and study (Fitch, 1987; Dorcas & Willson, 2009; Halstead, Wylie & Casazza, 2013). Our inability to detect snakes can be particularly problematic for species of conservation concern and for some invasive species, which can cause considerable ecological damage (Wiles et al., 2003; Engeman et al., 2011; Mayol et al., 2012; Reed et al., 2012; Piquet et al., 2021).

For both native and invasive snakes, there are two broad approaches used to find and capture them: actively searching (e.g., transects and road surveys (Rodda et al., 2007)) and passive aggregation (e.g., drift fence-traps, funnel traps and cover boards (Dorcas & Willson, 2009; Fitzgerald, 2012)). There are also a number of more specialized methods used to detect snakes, including infrared cameras (Avery et al., 2014; Neuharth et al., 2020), scent-detection dogs (Vice & Engeman, 2000), scout snakes (i.e. radio tagged snakes tracked to breeding aggregations; Smith et al., 2016), pheromone lures (Mason & Greene, 2001), and incentivized removal programs (e.g., rewards for snake removal; Guzy et al., 2023). However, identifying a method or combination of methods that provides adequate detection of many snake species remains challenging (Durso & Seigel, 2015; Clark, Clark & Siers, 2017; Maggs et al., 2019; Boback et al., 2020).

Efforts to detect snakes by luring them to a specific area often consider how snakes perceive their environment and search for prey. Snakes rely on thermal cues, movement (visual and vibrations), and chemical cues sensed via olfaction (receptors in the nose; Byerly, Robinson & Vieyra, 2010) or vomeronasal organs (receptors on the roof of mouth; Byerly, Robinson & Vieyra, 2010) to find prey (De Cock Buning, 1983; Glaudas et al., 2019). Snakes generally target areas with a relative abundance of cues (Secor, 1995; Madsen & Shine, 1996; Tutterow et al., 2021), which can be manipulated to attract them. For example, decaying rodent carcasses and chemicals have been used to lure and capture brown tree snakes (Boiga irregularis; Savarie & Clark, 2006).

One infrequently used approach to increase the detection and capture of snakes is live mammalian lures, often rodents. Mammals provide a combination of thermal, movement, and olfactory and/or vomeronasal cues (Askham, 1992; Rodda et al., 1992; Reed et al., 2011; Yackel Adams et al., 2019). Using them as lures has been shown to increase the capture of brown tree snakes (Rodda et al., 1992; Engeman & Vice, 2001), but was less effective than avian lures (Yackel Adams et al., 2019). Live rodents have also been tested for ambush hunters like Burmese pythons (Python bivittatus) (Emer et al., 2022) who aggregated near the traps rather than entering them (Reed et al., 2011). Consequently, the failure of some mammalian lures, may be due to trap avoidance rather than lack of attraction to the bait (Reed et al., 2011; Bartoszek et al., 2021). Accordingly, there may be potential for mammalian lures to increase snake detections without increasing live capture success.

The need to increase the detection of snakes is particularly acute in the Greater Everglades Ecosystem (hereafter ‘the Everglades’) of Florida, USA, where detecting and removing invasive Burmese pythons is a conservation priority (Guzy et al., 2023). These very large snakes have played a direct role in the precipitous decline of the region’s mammals (Dorcas et al., 2012; McCleery et al., 2015; Sovie et al., 2016), which has dramatically altered food webs (Taillie et al., 2021; McCampbell et al., 2023) and host-parasite dynamics (Miller et al., 2018, 2020; Burkett-Cadena et al., 2021). There is also a need to find efficient an effective way to detect and monitor the native snakes in the Everglades and other systems (Durso, Willson & Winne, 2011). While the specific detection rates of native snakes in the Everglades are not currently known, they are likely to be comparable to the community of snakes found in the freshwater wetlands of South Carolina, where detection probabilities range from 3–46% (Durso, Willson & Winne, 2011). In comparison, the probability of visually detecting a marked or telemetered Burmese python is <2% and the probability of detecting any python when present is <5% (Nafus, Mazzotti & Reed, 2020; Guzy et al., 2023). These reduced detection rates are a function of Burmese pythons’ crypsis and use of aquatic environments (Dorcas et al., 2017; Hunter et al., 2019). However, recent research suggests that time-lapse cameras may increase these probabilities of detection (Cove et al., 2023).

To determine if live mammalian lures could increase detection of differ types of snakes (i.e., Burmese pythons, larger native snakes, and all snakes—including larger snakes—that are not pythons) in the Everglades, we conducted a controlled experiment. We coupled three detection methods (visual surveys, python-detection dogs, and time-lapse cameras) with domesticated rabbits (Oryctolagus cuniculus) in secure, predator-proof pens (treatment pens) and paired control pens (pens with no rabbits) to test the efficacy of mammalian lures for snake detection. Due to the combined thermal, olfactory, and movement cues the rabbits provide, we predicted that live rabbit lures would increase detections of different categories of snakes and particularly larger snakes and Burmese pythons. We also predicted that time-lapse cameras would have the best chance of detecting larger snakes like Burmese pythons (Cove et al., 2023) and would be more cost effective than other methods. Finally, we predicted that python-detection dogs would detect more pythons than humans (Guzy et al., 2023).

Materials and Methods

Study area

We conducted this study within the Everglades of South Florida, USA (Fig. 1) from 3 May to 1 August, 2021. The Everglades is the largest sawgrass prairie on the planet and contains critical habitat for endemic and endangered animals and plants (Lodge, 2016). We tested mammalian lures in two distinct areas of the system, the C-4 Impoundment (C-4) and the Frog Pond Public Small Game Hunting Area (Frog Pond), in Miami-Dade County, Florida, USA (Fig. 1). Both sites border agricultural lands and urban development as well as Everglades National Park. Rodents were common on both sites (McCampbell et al., 2023) and the probability of large mammal presence increases roughly across a north to south gradient across the study area (Taillie et al., 2021). Burmese pythons have regularly been removed from both sites and the surrounding areas (EDDMapS, 2021). No python removal activities occurred in our study area for the duration of this study.

Figure 1 Location of study sites and individual pens in South Florida, United States.

The study sites are represented by green dots and individual pens are represented by black dots. Paired pens located in Frog Pond site (bottom right) on the L-31W levee and pens in C-4 site (top right) located along the levee and one pair in the property interior.

C-4 is a 168 ha block of wetland surrounded by levees near the northern edge of Everglades National Park (Fig. 1). This block is dominated by marl prairie interspersed with sawgrass (Cladium jamaicense), hardwood hammocks, wiregrass (Aristida stricta), pond apple (Annona glabra), soft rush (Juncus effusus), and swamp fern (Blechnum serrulatum). Frog Pond is a 2,106 ha strip of wetlands on the eastern side of Everglades National Park (Fig. 1). The accessible portion of this area was adjacent to a levee (L-31W) vegetated with Carolina willow (Salix caroliniana), soft rush, cattails (Typha spp.), poisonwood (Metopium toxiferum), gumbo limbo (Bursera simaruba), and sugarcane (Saccharum officinarum).

Snake community

Twenty-six native snake species have been recorded in Everglades National Park, adjacent to our study site (Meshaka, Loftus & Steiner, 2000). Of these, four are venomous (eastern coral snake (Micrurus fulvius)), Florida cottonmouth (Agkistrodon conanti), dusky pygmy rattlesnake (Sistrurus miliarius), and eastern diamondback rattlesnake (Crotalus adamanteus). The most common native species include black racer (Coluber constrictor), rough green snake (Opheodrys aestivus), eastern rat snake (Pantherophis alleghaniensis), eastern corn snake (P. guttatus), southern ringneck snake (Diadophis punctatus), eastern garter snake (Thamnophis sirtalis), brown watersnake (Nerodia taxispilota), peninsula ribbon snake (T. saurita), Florida brown snake (Storeria victa), and cottonmouth (Dalrymple et al., 1991). Snakes of conservation concern include the federally threatened eastern indigo snake (Drymarchon couperi). In addition to the Burmese python, there have been at least three other invasive snakes (Northern African rock python) (Python sebae), Brahminy blindsnake (Ramphotyphlops braminus), and boa constrictor (Boa constrictor) recorded within <50 km2 of our study sites (EDDMapS, 2021). For this study we separated snakes into three categories, pythons, larger native snakes capable of eating mammals as large as rabbits, and all snakes– including larger snakes-that are not pythons (hereafter, ‘pythons’, ‘larger snakes’, ‘other snakes’,). We categorized eastern diamondback rattlesnakes, cottonmouths, brown watersnakes (Nerodia taxispilota), eastern indigo snakes, eastern rat snakes, eastern corn snakes, and Florida kingsnakes (Lampropeltis getula) as larger snakes with an increased probability of being attracted to larger mammalian lures (i.e., rabbits).

Study design

To isolate the effects of a mammalian lure, we placed six paired treatments (i.e., treatments and controls) at the Frog Pond site and three paired treatments at the smaller C-4 site (Fig. 1). We cleared most vegetation within 2 m of each pen to facilitate the detection of snakes. We placed paired pens 95–105 m apart and randomly assigned treatments. Each pair of pens was separated by >405 m. Treatment and control pens were 1 m × 1.5 m and 0.7 m tall, constructed with a wood frame and ½-inch stainless-steel mesh to exclude snakes and other predators (Fig. 2). We obtained domesticated rabbits from Rivenzale Ridge rabbit breeders in Jacksonville, Florida, USA. We chose domestic rabbits as a lure because native marsh rabbits (Sylvilagus palustris) are common prey for many of the region’s native and invasive snakes (Allen & Neill, 1950; Chapman & Willner, 1981; McCleery et al., 2015; Guzy et al., 2023) and radio-tracking of native marsh rabbits led to python and rattlesnake detections (McCleery et al., 2015). However, these native rabbits are now exceedingly rare in the southern portion of the Everglades (Sovie et al., 2016; Taillie et al., 2021) and domesticated rabbits provide a viable and readily available alternative. We placed two rabbits in each treatment pen, separated them with a wooden partition, and provided each rabbit with a wooden box for additional cover. We visited and inspected the rabbits daily. During our daily visit we replenished the rabbits’ food and filled their two water receptacles. Once a week, we scored their body condition per our approved University of Florida Institute for Animal Care and Use Committee protocols (IACUC study #201910726). Rabbits receiving a score of two (thin) or that appeared ill were to be brought to a local veterinarian for evaluation and treatment. If their score dropped below two (very thin), the rabbit was to be removed from the study and either treated or euthanized by a veterinarian. At the end of the study 14 of the 18 rabbits were adopted as pets and the other four were returned to the breeder.

Figure 2 Control pen (without rabbits) and game cameras.

Game cameras and pen set up at C-4 site. Photo courtesy of Marina McCampbell. 13 July 2021. Kendall, Florida.

Game cameras

To detect snakes, we placed two Reconyx HyperFire 2 Professional Covert IR cameras (Reconyx, Holmen, WI) approximately 2 m from each pen, aimed at the pens horizontally (Fig. 2). We deployed cameras on a continuous time-lapse of 1-min intervals for 90 days (3 May 2021–1 August 2021). Time-lapse can be used to detect reptiles that often do not emit enough heat to trigger camera sensors (Hobbs, Brehme & Crowther, 2017) or move enough to trigger camera motion sensors (Yackel Adams et al., 2019). We changed the memory cards and batteries in each camera once per week. We manually reviewed all images for snakes, identifying them to family and lower taxonomic levels when picture quality allowed. Using digiKam (2020), an open-source digital photo management application, we manually reviewed images at a rate of 5,000–7,000 photos per hour and tagged the snakes identified in the photos. Four observers independently each reviewed 60,000–80,000 images every week and removed photos without vertebrates present. All photos containing snakes were reviewed and organized by the lead author prior to analyses. Unknown snake species were reviewed by two professional herpetologists. We considered detections to be independent when at least 1 h separated detections of the same species at a pen (Sollmann, 2018; Neuharth et al., 2020). Additionally, we used CamtrapR (Niedballa et al., 2016) to extract the metadata (e.g., species, time, date, etc.) from the tagged photos and to determine the timing and length of visits from pythons, larger snakes, and other snakes. Our camera trapping protocols were approved by the University of Florida Institute for Animal Care and Use Committee (IACUC study #201910726), and our cameras were placed in areas with restricted public access mitigating the need for additional ethics clearances.

Visual surveys

In addition to time-lapse game cameras, we conducted visual surveys for pythons, larger snakes and other snakes at each pen 5 days per week. Starting at a different pen each day, we visually surveyed in concentric circles expanding every 2 m until we reached a 10 m radius from the pen. Searches lasted approximately 8 min per pen. To prevent overlap in searching with the python detection dog team (see below) we conducted searches in the morning and early afternoon (07:00–15:00). The same two observers conducted all visual surveys for the duration of the study to reduce observer bias. Both observers were trained in snake identification by University of Florida and Florida Fish and Wildlife Conservation Commission herpetologists.

Python-detection dog surveys

We collaborated with the Florida Fish and Wildlife Conservation Commission python-detection dog team to conduct canine surveys for pythons. The team consisted of one handler, two dogs, and a biologist to record data and identify snakes. The dogs were trained to specifically detect Burmese pythons, first with towels from python enclosures, then with live pythons in bags at a training facility in the Everglades. The team started surveys at the pens and searched within a 40 m radius for 5 min. They searched each pen three times per week and started their searches at a different random pen each time. Surveys were conducted in the evening (18:00–20:15) when dogs and handlers could navigate by daylight and pythons were believed to be active (Whitney et al., 2021), potentially making them more likely to be near a lure.

Statistical analyses

First, sample size permitting, we compared the number of snakes recorded using each method (i.e., camera, python-detection dogs, and visual searches), by fitting a general linear model to a Poisson distribution using the lme4 package (Bates et al., 2015). Next, we compared treatments (i.e., rabbit lures and controls) across the different methods and categories of snakes (pythons, larger snakes, other snakes). For larger data sets (>50 detections) we used a generalized linear mixed model using the glmmTMB package (Brooks et al., 2017). We again used a Poisson distribution and considered each pair of pens as a grouping variable for the random effect. To estimate the predicted detections for treatment and control pens, we reported the incidence rate ratios (IRR; number of events per time) by exponentiating the model coefficients to determine the magnitude of difference in detection (“incidence”) of snakes between treatments (Hilbe, 2011). We interpreted IRRs as the probability of detecting snakes at treatment vs. control pens (Hilbe, 2011). IRRs close to or equal to 1 suggest no difference in detections between treatments. IRRs exceeding or less than 1 suggest an increased or decreased probability of detections for snakes at treatment pens. We evaluated the fit of our models by plotting residuals (DHARMa; Hartig, 2021). For smaller data sets (<50), to avoid overparameterizing, we used a Chi-square test of independence to determine if detections at treatment pens differed compared with control pens. Additionally, we used time-lapse camera data to evaluate the average amount of time that pythons, larger snakes and other snakes spent at each pen, we totaled the time snakes spent at pens, calculated the median, and compared differences between treatment and control pens using a Wilcoxon rank sum test to account for nonparametric data (Leon, 1998). All analyses were performed in program R (R Core Team, 2021, version 4.0.3) and graphics were created using ggplot2 (Wickham, 2016). Data and code are available from the Figshare repository (https://figshare.com/s/63b9b4d9236c12fd4430).

Costs per detection

We estimated the cost per snake detection by category (i.e., pythons, large snakes, and other snakes) for methods that detected snakes by dividing the costs (in USD) by the number of snakes detected. The expenses for each method included labor costs, mileage, and equipment and supplies. We calculated labor costs at $15 per hour and used the University of Florida’s milage rate ($0.445 per mile) to calculate travel expenses. We did not include housing into our calculation, as the costs vary widely and are not commonly provided as part of an employee’s compensation.

Results

We collected a total of 3,421,440 pictures from our cameras and recorded 5,307 pictures with snakes present. After filtering for independent detections, we recorded a total of 375 snakes from 12 species and four families (Table 1). Time-lapse game cameras detected almost 40 times more snakes (n = 375, treatment = 245, control = 130; Fig. 3) than visual surveys (n = 10). The most common other species detected at pens were black racers (n = 84), eastern ratsnakes (n = 33) and Burmese pythons (n = 22; Table 1). Of the 22 Burmese python detections, 21 were observed at treatment pens with rabbits (Fig. 3). We completed 63 visual surveys and detection dog surveys per pen. Across the two sites and 18 rabbit pens, the python-detection dogs did not find a python, and our visual surveys only detected 10 snakes (non-python species; treatment = 8, control = 2; Table 1).

Table 1 Classification of 375 snakes detected by cameras and 10 snakes detected by human search (in parentheses) by treatment (control-without rabbits and treatment-with rabbits) at the C-4 site and Frog Pond site from 3 May–1 August 2021.

			C-4	Frog pond		
Family	Scientific name	Common name	Control (n = 3)	Treatment (n = 3)	Control (n = 6)	Treatment (n = 6)	Total	
Pythonidae	Python bivittatus	Burmese python	0	14	1	7	22	
Colubridae	Coluber constrictor	Black racer	7	6 (1)	37	34 (2)	84 (3)	
Colubridae	Nerodia taxispilota	Brown watersnake	0	0	0	1	1	
Colubridae	Unknown	Unknown colubridae	4	8	18 (1)	33 (1)	63 (2)	
Elapidae	Micrurus fulvius	Coral snake	0	0	2	1	3	
Colubridae	Pantherophis guttatus	Corn snake	1	3 (3)	2	7	13 (3)	
Viperidae	Agkistrodon conanti	Cottonmouth	0	1	10	1	12	
Viperidae	Crotalus adamanteus	Eastern diamondback rattlesnake	0	0	0	2	2	
Colubridae	Lampropeltis getula	Florida kingsnake	1	0	0	0	1	
Colubridae	Nerodia fasciata pictiventris	Florida banded watersnake	1	1	1	5	8	
Colubridae	Thamnophis sirtalis	Garter snake	0	6	0	0	6	
Colubridae	Pantherophis alleghaniensis	Eastern rat snake	1	15 (1)	2 (1)	15	33 (2)	
Unknown	Unknown	Unknown snake	4	4	38	81	127	
Total			19 (0)	58 (5)	111 (2)	187 (3)	375 (10)	

Figure 3 Independent detections of different categories of snake from time-lapse game cameras.

Detections are categorized by the type of snake (Burmese python = Python, large native mammal-eating snakes = Large, all non-pythons = Other) and pen treatment (Baited Treatment with rabbits = grey, Unbaitated Control without rabbits = black). Field work was conducted in Greater Everglades Ecosystem from 3 May to 1 August 2021.

Using a Chi-square test, due to a smaller sample size of pythons (<50), we found that pythons were significantly more likely to be detected by time-lapse cameras at treatment pens with live lures compared to control pens (χ2 = 18.18, df = 1, p < 0.001). Pythons visited pens predominantly (>98% of pictures) from 21:00 to 8:45 h (Fig. 4) and stayed at treatment pens for a median of 20.0 min. Cameras at the control pen detected a single python present for 4 min. Time-lapse cameras at pens with rabbits were also 165% (IRR = 2.65, 95% CI [1.52–4.62]) more likely to detect larger snakes (β = 0.973, z-value = 3.42, p < 0.001), that spent more time at treatment pens (W = 250, p < 0.036), with a median of 8 min compared to a median of 2 min at control pens.

Figure 4 Histogram of Burmese python activity detected by cameras.

Activity was measured as the number of pictures recorded per hour (24-h clock) at baited treatment and unbaited control pens. Dashed and dotted lines represent the timing of visual and canine sampling, respectively. Data collected from 3 May to 1 August 2021.

Finally, timelapse-cameras at pens with rabbits were 74% (IRR = 1.74, 95% CI [1.40–2.16]) more likely to detect snakes other than pythons when compared to control pens (β = 0.552, z-value = 4.99, p < 0.001). Like larger snakes and pythons, other snakes spent significantly more time at treatment pens (W = 10,575, p < 0.001), with a median of 3 min compared to 1 min at control pens.

We estimated the cost per snake detection by category (python, large, other) for both the methods that detected snakes, visual surveys and time-lapse cameras. The overall cost to conduct rabbit-lure camera surveys was $55,242. These costs came from the labor needed to care for rabbits, monitor cameras, tag and organize photos, and drive to sites ($17,565). Additional costs included pen construction ($14,400), purchasing rabbits and rabbit supplies ($2,721), cameras ($15,260), batteries and digital storage ($703), as well as milage for travel to and from field sites ($4,593). The total cost to conduct the visual surveys was $30,954. The costs for this portion of the project included all rabbit related expenses and the labor needed to survey each pen five times per week ($2,835). However, this portion of the project did not include the expenses of the cameras themselves nor the costs associated with storing and sorting images. For time-lapse cameras the cost per python ($2,511), larger snakes ($891) and other snakes ($156) was considerably reduced when compared to the cost per detection for visual surveys (python = no detections; larger snakes = $6,189; other snakes = $3,095).

Discussion

Our study presents compelling experimental evidence that live mammalian lures coupled with time-lapse cameras can improve the detection of snakes. This combination of methods increased overall snake detection by 74% at treatment pens compared to controls and greatly enhanced the detection of larger snakes and Burmese pythons. When detected by time-lapse cameras all three categories of snake spent more time at treatment than controls.

Cameras are increasingly being used by researchers to detect squamates such as snakes and lizards (Welbourne et al., 2017; Neuharth et al., 2020; Ryberg et al., 2021; Walkup et al., 2023), likely because of their ability to collect continuous data. This functionality can be especially useful for species that are active nocturnally, like Burmese pythons were in this study (Fig. 4). An additional, advantage of time-lapse cameras is that they can provide some information on behaviors. For example, while it was not the focus of this study, we did find evidence of pythons actively investigating and climbing on pens with rabbits (Fig. 5).

Figure 5 Examples of python pictures at rabbit pens.

Picture shows four different examples of pythons investigating pens with domestic rabbits.

Python-detection dogs and visual surveys have had some limited success in other studies in the Everglades (Dorcas et al., 2017; Guzy et al., 2023), but we found that they were not as efficient and effective as time-lapse cameras at detecting snakes. Based on the activity periods of pythons in this study, it appears unlikely that pythons would have been actively seeking lures during the day light hours that were logistically feasible for python-detection dog and visual surveys. These approaches could possibly be more effective during the winter and spring when other research has suggested pythons are more active diurnally (Whitney et al., 2021; Cove et al., 2023). Another potential shortcoming of searching for snakes with humans and dogs in this region is that they may be constrained by seasonal flooding (Cablk et al., 2008), safety concerns (e.g., alligators, venomous snakes), and thick vegetation (Dorcas et al., 2017).

While results clearly show the advantages of time-lapse cameras coupled with live mammalian lures, there were several constraints that limited the comparison of our different approaches to detecting snakes. First, each method searched different areas around the pens (cameras ≈ 4 m, visual searches = 10 m, dogs = 40 m). Instead of focusing on a standard area, we chose to maximize what was logistically feasible and could be replicated by other researchers and managers given time, budgetary and safety constraints. For example, because detection dogs search areas more quickly than humans, we let them search a 40 m radius. Expanding visual searches to this area would have reduced the frequency of our sampling and expanding the cameras detection area to 40 m would have made the cost of cameras and picture sorting prohibitively expensive. Moreover, given that there were no detections by dogs and minimal detections from visual searches, there does not appear to be a need to adjust our data for standard, yet unrealistic sampling areas. Another issue that was difficult to avoid was temporal bias. Due to safety concerns and the terrain of our study sites, it was not advisable to have humans or dogs sampling at night. While visual searches would have likely been limited by low visibility, it is possible the dogs working under safe nocturnal conditions may have had a better chance of detecting snakes. Finally, because the Florida Fish and Wildlife Conservation Commission python-detection dogs were only trained on pythons, we could not assess the ability of detection dogs to find snakes that were not pythons.

While using time-lapse cameras was the most cost-effective method for detecting pythons and larger snakes, this approach is constrained by the manual detection of snakes in photographs. This shortcoming can likely be addressed with the implementation of automated photo sorting programs that are commonly used for camera trapping of mammals (e.g., Swinnen et al., 2014; Price Tack et al., 2016). While most programs are optimized for detecting larger mammals, there are generalized Artificial Intelligence (AI) image processors such as Megadetector v5.0 (Beery, Morris & Yang, 2019) and others (Bolon et al., 2022) that have been trained with images of reptiles. These programs automatically detect photos with animals by estimating the change in pixels between time-lapse photos (Swinnen et al., 2014) and can reduce time spent reviewing photos. These methods are most effective when deployed in areas with homogenous backgrounds that do not interfere with pixel comparisons (Swinnen et al., 2014). Image processors can struggle to identify smaller snake species (<50 cm), but this shortcoming could likely be improved by reducing the camera focal length, standardizing backgrounds, and including a scale (McCleery et al., 2014). These improved efficiencies would likely only decrease cost by 6–8% in a onetime deployment; however, after the initial purchase of cameras and rabbit pens, automating photo processing could likely reduce the cost per detection by >30%, due to reduced labor costs.

Like the collection and processing of images, finding alternatives to live lures would greatly reduce the cost and animal husbandry needs. There are several potential options for inanimate lures that might be able to attract snakes without the welfare and monetary constraints of live lures. Some snakes rely heavily on chemical cues (Cooper, 1991) to locate prey. Accordingly, scents derived from domestic rabbits in the form of feces, urine, and hair may provide a viable alternative to live lures. As an example, Worthington-Hill, Yarnell & Gentle (2014) found no difference between corn snakes’ attraction to a live mouse lure and the scent from soiled mouse bedding. Snakes also sense and can be attracted to heat and its infrared radiation (Gracheva et al., 2010; Bakken et al., 2018). Thus, it may be possible to attract snakes from heat sources that emit heat in the range of their prey (Bakken et al., 2018). Moving and vibrating prey replicas (Worthington-Hill, Yarnell & Gentle, 2014) may also attract snakes (Haverly & Kardong, 1996), as movement and vibrations (Young & Morain, 2002) are an important means for snakes to identify prey. Finally, there is some evidence that suggests prey cues from inanimate objects are more effective snake lures when they are used together (Shivik, 1998).

Similar to the findings in this study, live and dead lures have been an effective method for attracting arboreal (Rodda & Fritts, 1992; Rodda et al., 1999) and aquatic (Keck, 1994; Winne, 2005) snakes. However, looking at the efficacy of live rodent lures to capture brown tree snakes, Walkup et al. (2024) suggested that lures maybe less effective when prey is more abundant. While it is possible that prey abundances might alter trap efficacy, rodents, the most common prey of the of Burmese python (Guzy et al., 2023) and larger snakes in our study, are still common at both our sites (McCampbell et al., 2023). Moreover, we detected more pythons at the northern extent of our study (i.e., C4) where more larger mammals have been detected (Taillie et al., 2021).

Conclusions

We clearly show that the detection of larger snakes can be enhanced with mammalian lures and time-lapse cameras. Unfortunately, the use of cameras and manual review of images did not allow for the physical capture of snakes. However, it may be possible to adjust the current design to facilitate capture. Future designs should consider game cameras that are connected to cellular networks (Nazir et al., 2017) and notify researchers when they are triggered by pressure plates (Swann, Kawanishi & Palmer, 2011) or AI algorithms that isolate snakes or target species (Staab et al., 2021; Roy et al., 2023). Alternatively, installing funnel traps with a one-way door (i.e., Reed et al., 2011) may provide an opportunity to capture snakes that regularly investigated rabbit pens (Fig. 5). Importantly, the inability to physically capture snakes does not preclude the ability to conduct rigorous population studies. Machine learning programs have been trained to recognize unique coloration and spot patterns of individual snakes (Yang et al., 2013; Phon-Amnuaisuk, Au & Omar, 2016). Researchers could then potentially use capture-mark-recapture datasets to estimate population parameters of larger snakes and fill critical knowledge gaps to develop more effective conservation and management strategies.

Supplemental Information

Supplemental Information 1 Snake3 dataset.

Supplemental Information 2 Timestamp Raw Data.

The only variable listed is Time. This represents the time a picture of a python was taken on a 24:00 time scale.

Supplemental Information 3 Raw Data.

Supplemental Information 4 Code.

Additional Information and Declarations

Competing Interests

Author Contributions

Animal Ethics

Data Availability

McKayla M. Spencer is employed by Florida Fish and Wildlife Conservation Commission, and Kristen Hart is employed by United States Geological Survey.

Marina McCampbell performed the experiments, analyzed the data, prepared figures and/or tables, authored or reviewed drafts of the article, and approved the final draft.

McKayla Spencer analyzed the data, authored or reviewed drafts of the article, and approved the final draft.

Kristen Hart conceived and designed the experiments, authored or reviewed drafts of the article, and approved the final draft.

Gabrielle Link performed the experiments, authored or reviewed drafts of the article, and approved the final draft.

Andrew Watson analyzed the data, prepared figures and/or tables, authored or reviewed drafts of the article, and approved the final draft.

Robert McCleery conceived and designed the experiments, analyzed the data, prepared figures and/or tables, authored or reviewed drafts of the article, and approved the final draft.

The following information was supplied relating to ethical approvals (i.e., approving body and any reference numbers):

The University of Florida Institutional Animal Care and Use Committee approved the study (IACUC study #201910726).

The following information was supplied regarding data availability:

The code and datasets are available in the Supplemental Files and at figshare: McCleery, Robert (2024). Mammalian lures monitored with time-lapse cameras increases detections of pythons and other snakes. figshare. Dataset. https://doi.org/10.6084/m9.figshare.25834828.v1.

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
