# Peer review of "Mammalian lures monitored with time-lapse cameras increase detection of pythons and other snakes"

_PeerJ, doi:10.7717/peerj.17577_

## Round 0.1 · original submission · Major Revisions

I really like the idea behind this study. I think it is a useful concept, and I think you ran a good experiment. In fact, many years ago, after I first joined some Fish and Wildlife folks and people from the park on a python survey, I returned to my undergraduate university and pitched several mammal lure, python trapping methods studies to my profs. These ideas never got off the drawing board. So needless to say, I am excited to read what you did and what you found. Great work on a useful and interesting study.

The Reviewers are in agreement, and they too see the value of this work, and have provided some great comments to help you present your research in a way that will maximise its impact. Similarly, I have provided comments and suggestions as well (see below). Both of the Reviewers’ comments, and those of my own, are all aimed to help you continue to improve this manuscript with the intent to shape it into a form whereby it can reach the most people - who will read it, understand it, and potentially use it in the future; as well as use this work to direct future research on this topic.

Given that some of the suggestions and recommendation include the addition of some extra analysis and some reframing of this studies narrative, I have recommended that it receive the decision of asking for “Major Revisions”. You will see the Reviewers both suggested “Minor Revisions”, however given the number of changes that are suggested – not overly difficult ones, just widespread and numerous – I would really like you to take the time to really work them in. From a general science standpoint, I totally agree, Minor Revisions (like what you did was pretty good). But from a study presentation and manuscript perspective, this is where some more work can bring the level of writing up to a standard that does your research the service it deserves. The research was great, now we want the writing to do it justice and let this study shine.

I hope you find our comments, suggestions, and recommendations helpful as you revise this manuscript. I am very much looking forward to reading the next version.

General Comments
Across the manuscript the writing could use quite a bit of improvements. There were quite a few typos, areas of awkward phrasing, and aspects the sentences and paragraph structure that could use some more polishing. This is not to criticize your abilities as writers – we can always improve our writing – but merely to note that such improvements are needed to provide this wonderful study with the writing prowess it deserves, and to make it so readers of the paper are able to flow through your narrative and gain insight from your findings. The Reviewers and myself have caught a good deal, but I am certain there are more. So be vigilant. Also, and all to frequently, statements are missing some deeper context. Remember that we want our writing to provide the reader with all of the information they need -both when they are going into a study so they have the context they need to follow you, and in the Discussion especially so they can understand the gravity and reasoning for what you found. Across the manuscript there was a tendency to be somewhat overly succinct, a little bit is good, but too much does the readers a disservice and increased the chances of them getting confused or lost.

One suggestion to improve in this area is simply to view each and every paragraph as a standalone idea. Ensure it begins with a strong and clear topic sentence that explains what the purpose of the paragraph is, then follow it up by making several assertions about the topic, followed by several examples of how this fits into what is already known, (and/or what is not known about it – in the Discussion this could be direction for future work in the middle paragraphs), and end the paragraph with a summary of the ideas previously covered and a segue/transition to the next topic that will be presented in the following paragraph. This makes for a more contain set of well supported ideas, as well as guiding the reader through your thought process. In short, it helps give our writing more structure and flow.

Lastly, and as you revise, look for areas to continue to deepen your readers understanding. We need to assume not everyone reading this is well versed in snake trapping or your study system. I genuinely would love to see the Introduction and Discussion increase by 50-75% each, not by adding more concepts, but by simply expanding on the ideas you present and give them more context and content, so it is easier for readers to follow along. Try to imagine they are well educated, but new to this field. When you read “scout snakes” or “mammalian lure” or incentivised removal program” what else would you need to know to be able to follow along? Or when you are explaining how trap efficiency is increase by exploiting a target species’ natural history and/or biology, what sort of theoretical background should the reader have?

Major Comment:
1) I was happy to see that Reviewer 1 brought up the idea of incorporating a cost-benefit analysis into this paper, as it was one of the first things I noticed was missing when I conducted my first read through of this manuscript. Given that the idea of the paper is to speak to the efficacy of mammalian lures as a methods, and you conducted alternative methods (surveys, dogs, and cameras with no lure), it seem like the logical way to present your data as well. So not to replace the analysis you have but rather, add another layer to this study. Surely, you are able to calculate the cost of running the surveys (for people) and people with dogs, as well as the cost of cameras and cameras when rabbits and their care was added. Then you have a dollar value and a # of snake seen per method. This would allow you to expand your discussion about how lure may be able to better optimise search effort. I strongly suggest this is a worthwhile aspect to include in the next version.

2) What was the original intent of this work? I do not ask this to be rude or accusatory. In reading this, and some of the Reviewer comments point to this idea/feeling as well, it feels like this was originally planned as a python detection study, and the collection of other snake data came in as a bonus… and then when you went to write it, you decided to include the other snake data to widen the breadth of how this lure method could be used. In the end, I agree talking about the others snake seen is has absolute merit. But if this was a planned python study, which then also brought in some neat other snake species insights, then it should be presented in that way. If this was always intent on being a multi-species study from the onset (which if so, then that’s cool), but then I think the introduction should be expanded some more to provide a fuller grounding, so this ambiguity does not arise. But as I said, it is not how it feels as you read how you currently are framing it, nor in how you planned the methods. Like the choice of rabbits was clearly for pythons, rather than for something like cottonmouths or garters. So, I think you can still frame this as the original study may have been intended, but then include the additional information about other species encountered and how this method could be used for native species surveys as well. In the end, the point I am making is about transparency. If this was conducted as a python study and other fascinating insights were also gain, it should be presented as such, rather than if it was always meant to be an all-snakes study. This will change a bit of your framing, but ultimately not the actual methods, or results, or even much of the Discussion.

3) Although I really like the concept of this study design, there are some clear issues with compatibility due to spatial and temporal factors that were not standardized. Given the outcomes of the study, I do not think these are fatal flaws for the work - which is a relief – but they do very much need to be owned up to and discussed, either in the methods with justifications for why it is acceptable. Or, and this is what I would suggest, discussed in the Discussion as a limitation and something that could be improved upon for future research on the topics (which I really hope this paper will spur). One of the issues is that your visual surveys covered 10 m from the pen, but the dog survey was 40 m – so that’s not directly comparable. Why did the dog team cover so much more ground than the human only team? Logically for a better comparison, you would have looked at how much area the camera covers and done both the in-person survey types over the same area. The other issue is the dog surveys happened at night (when you expected the snakes to be more active) and the visuals in the morning and afternoon. This is a clear temporal bias. Again, ideally you would have either randomised the times for each, or (and more logistically) alternated to balance them. The cameras were on night and day, so night and day visual surveys and night and day dog surveys would be the appropriate study design. Again, I know this cannot be changed now, and given your results, and the order it was done, it is not a make-or-break issue. But nonetheless this flaw in the methods needs to be discuss as to why you believe it is not a serious issue, and then recommendations made for how studies like yours in the future could be improved upon.

4) Staying in the same vein of the original intent for this paper. The way the methods are set up are as a test of four different detection methods (visual surveys, scent dogs, cameras, and cameras with a mammal lure. Surely when you were designing this you had an idea about which one would be best (the cameras with a rabbit pen; which you do state, which is great)… so this is your hypothesis to the question “which method would be best to detect pythons”. So, your prediction would be that due to the hunting behaviour of these snakes and their relatively cryptic nature, luring them to a constantly snapping camera trap would have the best change at logging detection. But you would likely also have predictions, and no doubt did while planning, about how the other methods would also fair. Camera trap with no bait vs the visual or dog surveys. Visual vs dogs. All of the ideas that I am sure you thought of went designing this great experiment can be discussed in the final paragraph of the introduction to set up the paper in the stronger way by setting out your hypothesis and some clear predictions. You can lay out some of the differences in detection based on previous studies, and how the unique aspect of the python invasion in Florida could make detection even more of a challenges… so testing which one works best is critical (which would be expanded on in the body of the Introduction), then end the Intro by posing your questions (which one would work best), your hypothesis (the method that keeps the snake in the area of the survey method the longest, thus maximising detectability), and then the predictions (like camera trap with mammal lure will see the most, followed by x, y, and z). I am not suggesting you create a hypothesis framework after the fact, but surely when you were planning it, you had reasons why you picked these four methods, and likely hunches for success (predictions as to which one would be best and which one the least). I personally would have predicted dogs would score much higher, and trained expert people too. So, if you were in a same boat, then be honest – it is ok for our predictions to be off if it is what we were thinking when we planned the study. Once again, the reason I am writing this is to increase openness and transparency in the reasoning for the study, and what your hypothesis and predictions were going into it. This makes for not only more open and honest writing, but in framing our thought processes in a hypothesis-driven framework we provide more structure to the paper. We often discount the fact we are hypothesis testing when designed any experiment – maybe because we did not explicitly write it down. But in creating this study you had reasons for picking these methods, and likely hunches on how’d they’d fair, and reasons for those hunches… this is what I am suggesting you share with the readers; in a somewhat more formal way. I want to be clear, I am not suggesting a HARK-ing approach here, rather, I am suggesting you go back into your notes or recollections of the planning phase of this work and include some information to give the readers some better context and this paper some better structure.

Specific Comments:
Line 65: May want to merge the two first sentences, or rephrase the topic sentence to mention detection approaches, as this is half of the paragraph’s content.

Line 72: replace the comma with a colon after “snakes”.

Line 73: There are no citations for the “transect or road surveys” statement

Line 78: Replace “finding” with identifying or quantifying. The problem is not finding a method, but knowing what the ‘best’ method is.

Line 80: Before jumping into the next paragraph, which is focused on mammal lures, I think it is worth while having a paragraph in between that discusses the concept of effective trapping relating to methods that capitalise on the movement patterns or sensory systems of the target species. Think of it as in “trap design theory”, provide some framework for the readers to understand why something like a mammal lure would be expected to work. Here you could talk about variety of traps, from fish to insect, to reptile, to mammal, that use/exploit aspects of the target species biology and ecology to increase trap effectiveness.

Line 81-82: Try not to bury to subject of a sentence, instead lead with it. Here you could flip this sentence to say “Finding methodological approaches that enhance snake detection is of growing importance to the management and conservation of snake populations.”

Lines 82-84: Provide some more context for what you mean by “live mammal lure”. Give a definition and in the next few sentences, maybe even a clear example (like with the rodent studies).

Line 86: Merge parentheses (Boiga irregularis; Rodda, 1992; Engeman & Vice, 2001)

Line 96: Make the clauses parallel- “The Everglades’ vast aquatic nature and the Burmese python’s crypsis”.

Line 98: add a qualifier like “in this environment” or “in this invasive range”, the Everglade’s vast aquatic nature does not make the Burmese python difficult to detect in other habitats. Also, the use of “notoriously” here is a bit to colloquial and colourful for scientific writing. Keep an eye out for this across the manuscript, not just for this word, but others that fit the same sort of vibe. Here, the same statement can be made to plainly say, that the nature of the size and structure of the Everglades environment makes this highly cryptic snake exceedingly difficult to detect. The term notorious refers to something being famous or well known, typically for something negative. So we should not be assuming that every reader of this article is familiar with this snake species or likelihood of finding them in the Everglades.

Lines 109-113: I think there is real value in framing why rabbits were used, and I totally agree you made the write call. But this information belongs in the Methods section when you are presenting your trap description, rather than in the Introduction – especially in this last part of the Intro where you are setting up your hypothesis and predictions.

Line 118: Add comma after “August”

Line 119: Replace “include” with “contains”

Line 123: Either remove the “and”, or change agriculture to “agricultural land”

Line 149: Remove “also”

Lines 154-172: This is a good place to explain why rabbits were used.

Line 163: Remove the ‘s’ from pens

Lines 194-201: Include some information about the surveyors. Are these snake experts? If so (great), are they well-trained with native species and pythons, just natives, or just pythons?

Lines 201-202: Was there a bias as observers becoming more proficient at spotting snakes/became more familiar with the survey route and area? If they are already expert level snake searcher then maybe not, but some context here would be good.

Lines 203-211: The dog team need to have more context given. How were they trained and for what purpose. Like if this is supposed to be an all snakes survey, were they trained to find all snake scents. If they were trained to detect python scent, this should be explicitly stated. This could be a form of bias, if the camera captures snakes equally, the visuals detect snakes equally, but the dogs are train to focus specifically for pythons. Given the dogs found nothing, this is not surprising if they were only trained on pythons, and during the 5 min period they were there, a python was not present. But this all should be openly discussed.

Line 210: Typo? ‘Potential’ is in this sentence twice, rephrase the second half of the sentence as “and thus expected to be more likely to be near a lure”

Line 221: Naming the R platform is redundant here, as in Lines: 237-238 you state that “All analyses were performed in program R”.

Line 239: A link to the Figshare Repository should be included here

Line 242-243: Make the clauses parallel- “our scent dogs did not detect any snakes, and our visual surveys only detected….”

Line 244: Add an ‘s’ at the end of camera, also avoid including any interpretation of the findings in the Results section (i.e., “… were clearly superior…”). Save that for the Discussion. Here just tell us exact what was found (e.g., values, counts, and model outputs).

Line 248: Add a comma after August. Also, this all belongs in the Methods section rather than the Results section. Remember the results are for what you tested and found. You were not testing when you deployed the cameras, or how many pictures it could take (which holy moly 3.4 million!). Those belong in the Methods. Here, you can plainly state that the cameras recorded 5,307 snake pictures, which results in the detection of 375 individual snakes.

Lines 265-266: The fate of the rabbits is not a stat or model outcome, thus is does not belong in the Results. One suggestion would be to subheading in your data collection section of the Methods, where you discuss the animal ethics approval, the research permission, the fate of the rabbits, and the care for the search dogs. This subheading could come after the subheadings within a “Data Collections” section (which would sit between the “Study System” and “Data Analysis” sections), so you’d have “Study Design”, “Camera Trapping” (with and without lures), “Visual Survey”, “Dog Surveys”, and “Ethics and Permission, and Animal Care”

Line 272: Again, remove notoriously and replace with a more appropriate phrasing for professional writing.

Line 274: These behaviours caught on camera are worth the reader seeing in the main paper (rather than just in the ESM), maybe make a composite figure with a handful of these observation and include them in the Results when you are presenting your camera trap findings. Like, the cameras detected pythons (check), also other snakes (excellent bonus), and we gained knowledge of the reasons the snake were at the traps by noting behaviour. You can even include a breakdown of the behaviours that were seen by the individual 22 pythons (what % were coiled, or sitting on the cage or striking, etc.).

Line 276-285: I think you really should expand this point. You study found mammal lures bolster detection rate by quite a margin. Which is excellent. But you rightly bring up the welfare issue. If rabbit scent is all that is needed, then soiled bedding could serve the same purpose and be less labour intensive for the people, less stressful for the animals, and less costly from a monetary perspective. The equal effectiveness of live rodent to rodent scent was that seen in one of the studies you cite. This study and the findings of rodents and rodent scent as a mammal lure should be explain in more detail here as well. So, instead of merely touching on this concept and moving on, take time to explain why and give context to the reader. This is based on the sensory systems of the snake and its hunting strategy. Could using soiled rabbit bedding be an equally effective method (maybe, it needs to be tested), but I would advocate you explain how it could be more useful from a management standpoint; as caring for a rabbit colony in one place (like a temp controlled building) would be way easier and likely cheaper, and then the field aspect is just regularly replacing the bedding pile at camera sites. Bedding could also be place withing actual traps that confine and hold the snakes that enter them. I know these ideas are not the direct purpose of this exact study, but they are the next logical progression of your work, and thus here is where there is some real value in talking about it to guide future effort.

Line 281: Place “or in combination” within a set of commas.

Line 292: Add a comma after “study”

·

Basic reporting

The reporting is clear, unambiguous, and follows a logical path.

Experimental design

This is simple and clear with appropriate detail.

Validity of the findings

The findings are clear and unambiguous.

Additional comments

It seems unlikely that snake traps will ever eradicate the invasive Burmese pythons from the Everglades, but the hunt for a cheap and efficient trapping method is important, as it could potentially lower the total python numbers and reduce the pressure on their native prey species. This MS clearly started out to test the combination of live rabbits as lures for attracting, and time lapse photography for recording invasive pythons, but morphed into a more general “all snakes’ MS.
This MS should be published, as it is a step towards that end goal, and shows unequivocally that live rabbits do attract pythons (and at least one native snake species).
The inclusion of some cost-benefit discussion would be beneficial, as this was clearly an expensive research project due to (a) the use of live rabbits (necessitating 90 daily visits to each rabbit), and (b), manual sorting of 3.4 million photos. The authors briefly discuss alternatives to both of these expensive problems, but I would like to see more discussion of the possible alternatives to live rabbits.
And please add some recommendations for future python lure studies that do not use live mammals: More discussion on the importance of scent, visual, and vibration lures to attract pythons would be helpful. For the corn snake Worthington-Hill et al. (2014) found no difference between a live mouse lure and (presumably fresh) “soiled mouse bedding”. Would a python trap using dirty rabbit bedding (from Rivenzale Ridge rabbit farm perhaps?) as a lure, replaced perhaps once or twice a week, work equally well? Would the addition of a solar battery-powered random movement and vibration device at night increase trapping success? These could easily be tested in future studies, perhaps even in large captive enclosures.
What percentage of python visits to rabbit lures would have been captured by, say, one photo every 30 or 60 minutes, as pythons stayed at rabbit pens on average 81 minutes? Am I right is thinking that no python appears to have (a) stayed more than about 2 hours near a rabbit lure, and thus none coiled and waited in the typical python ambush position, or (b) returned to the rabbit lure on the following night? Please make this clear in the Discussion.
Fig 3: Eyeballing this data, the increase in snake sightings at live lures is mostly due to the 98 eastern rat snakes and the ‘unknown colubrids’ that were clearly attracted to the rabbits. The other 10 snake species are seemingly not attracted to rabbits. Is this enough reason to use “snake’ in the title, rather than ‘Burmese python?” I will leave that one to the Editor to decide.

Other changes required:
Line 52, Abstract: Fig 3, line 253 and the raw data file all show that 21 pythons were recorded at rabbit lures, 1 at the control, however the Abstract (line 52) states 22 pythons visited rabbit lures, 1 at the control. The abstract is clearly wrong.
Line 60 should say: For larger snakes ‘with individually distinctive patterns’, this method….
Line 164 should be: each ‘with a’ rabbit box…
Line 163: pens should be ‘pen’
Line 210-211: Many typos so it’s a nonsensical sentence.
Line 296: What is a ‘solution hole’? Need to explain, use other terminology or just delete.

·

Basic reporting

Overall adequate. Two things:

I was blown away that you were able to go through 3.4 million photos. How did you do this? In the acknowledgments you thank two people for organizing “thousands” of photos but you collected millions. You need to include more detail about your process for sorting through the images and identifying which ones contained snakes, as well as your process for identifying those snakes to species.

Second, what were most snakes doing? Climbing on the cage? Even with cleared vegetation, it seems impressive that cameras could detect snakes at 2 meters. Can you categorize the snake images by behavior? I don't think this is essential but I was interested to know more.

Experimental design

Paired control/treatment enclosures are a strong design, I see no reason to doubt elements of the experimental design.

Validity of the findings

My most important comment is that these findings might only be generalizable to places where mammal population densities have declined by >90%. The attractiveness of mammalian baits might be significantly reduced in ecosystems where mammal-eating snakes are less prey-limited.

You should also be clear that your method and your findings would be expected to apply only to snakes that eat mammals. Thus, statements like “we predicted that live rabbit lures would increase detection of Burmese pythons and all other snakes” (lines 107-109) are misleading — I wouldn’t predict that rabbit lures would increase detection of Opheodrys, Thamnophis, Storeria or Diadophis, because snakes in these genera do not consume mammals or are too small to eat rabbits.

Additional comments

Here are more minor comments and suggestions:
Title: change “increases” to “increase”
Line 43: add “s” to the end of “provide”
Line 61: add “potentially” before “allowing researchers”
Line 70: omit symbols before “cause”
Line 80: you might be interested in the short review of this topic in Journal of Herpetology (49:503-506).
Line 83: insert “mammal-eating” before “snakes”
Text on lines 91-96 is largely redundant with that on lines 105-109, I would essentially omit lines 91-96 and/or combine with lines 105-109.
Lines 96-97: Rephrase as “The vast aquatic nature of the Everglades and crypsis of Burmese pythons…”
Line 106: insert “time-lapse” before “cameras”
Lines 109-111: presumably domestic rabbits are also easier to obtain and there are fewer human health risks than using marsh rabbits
Line 140: lowercase “e” on “eastern coralsnake”
Line 144: Pantherophis guttatus (guttata is an older, incorrect spelling)
Line 146: Thamnophis saurita (sauritus is an older, incorrect spelling)
Line 150: prefer “blindsnake” one word
Line 151: scientific name is Boa constrictor, not “Constrictor constrictor”
Line 151: replace “in” with “within 50 km2”
Line 159: insert “by” after “separated”
Line 162: omit “a” before “Rivenzale” and insert “Florida” after “Jacksonville”
Lines 162-164: Reformat as “We placed 2 rabbits in each treatment pen, separated them with a wooden partition, and provided each with a rabbit box.”
Line 165: I would suggest either including the specific brand or omitting the parenthetical
Lines 167-168: Omit “For our approved animal husbandry protocols”
Line 169: insert “a” before “local”
Line 175-177: make it clear that the cameras were aimed horizontally rather than vertically
Line 178: I presume another advantage of the time-lapse is that you got fewer pictures of rabbits
Lines 210-211: rephrase as “…were believed to be more active (>1800 h; Whitney et al. 2021) and potentially more likely to be found near a lure.”
Line 217: lowercase “l” on “lme4”
Line 244: add “s” to “camera”
Line 252: reformat as “…were black racers (n = 84), eastern ratsnakes (n = 33), and Burmese pythons (n = 22; Table 1)”
Line 260: Insert “Cameras at” at beginning of sentence.
Lines 264-265: consistently abbreviate “minutes” as “min”
Lines 265-266: Omit last sentence, already stated in methods
Line 272: rephrase as “detection of notoriously cryptic Burmese pythons”
Lines 283-285: I would suggest rephrasing this as something like “…live mammalian lures may provide researchers with the greatest amount of snake detection augmentation.”
Lines 289-290: See also Walkup et al. 2022. Testing the detection of large, secretive snakes using camera traps. Wildlife Society Bulletin:e1408; Ryberg et al. 2021. Effective camera trap snake surveys at a rarely accessible longleaf pine savanna. Herpetological Review 52:719-724; Amburgey et al. 2021. Evaluation of camera trap-based abundance estimators for unmarked populations. Ecological Applications:e02410.
Line 305: See Bolon et al. 2022. An artificial intelligence model to identify snakes from across the world: Opportunities and challenges for global health and herpetology. PLoS Neglected Tropical Diseases 16:e0010647.
Line 309: You should make the distinction between differentiating a snake from background and identifying species of snakes in images. There are computer vision algorithms that can do both, but they are distinct tasks. In your case, it seems that the first would be most useful.
Line 315: I’m not sure that pythons are “the most elusive” snakes; at least they are big. I would suggest that small fossorial snakes like Tantilla are even more difficult to detect than pythons, especially using camera traps.
Lines 327-328: The algorithm in Durso et al. 2021 does not differentiate among individual snakes, only among species.
Lines 328-330: I’m not aware that camera-trap-based CMR has been validated for snakes, but my impression is that it might work but if reducing density is the goal it’s likely preferable to just install one-way funnel traps and actually capture the snakes that are attracted to the lures.
Check alignment of indents, page breaks, and sentence-case on article titles in reference section
Line 457: correct formatting of scientific name “Boiga irregularis” (italics, lowercase “i”)
Figure 3: Move rabbit icon below “treatment” so that it doesn’t appear that the black bars are the rabbit pens.
Figure 4: Can you show using vertical lines the times of day when visual surveys and dog surveys were done?
Table 1:
In southern Florida, neither Nerodia fasciata fasciata nor Nerodia fasciata confluens occur (only N. fasciata pictiventris). What led you to think you had images of the other two subspecies and why did you separate them? If you are sure all the images are of N. fasciata, I would suggest combining them at the species level.
Burmese pythons are now known as Python bivittatus and are no longer considered to be a subspecies of Python molurus.
In Florida, cottonmouths are now Agkistrodon conanti, not A. piscivorus (as you’ve done on line 141).
Kingsnakes in southern Florida are Lampropeltis getula, L. floridana is not considered valid because of smooth intergradation of genes and phenotypes.
Have you tried posting the “unknown snake” and “unknown colubrid” images to a citizen science platform like iNaturalist or Zooniverse to crowd-source their possible ID?
I would like to see the supplementary figure with example images in the main manuscript.

---

## Round 0.2 · Minor Revisions

The authors did a great job addressing the comments and ideas brought forward after the first round of reviews, and I (as well as the reviewers) very much feel like this manuscript is polishing up nicely. Great job! One of the reviewers was able to catch a few minor points (see Reviewer 1's comments), but overall feels this manuscript is 'ready to go'.

Reviewer 2 also expressed that this manuscript is much stronger (as a publication) than the previous version, but in reviewing this current draft caught a number of important details and issues that I think are well worth addressing before we proceed with accepting this manuscript. Notably, there are a few citations that may have been misinterpreted and should definitely be shored up (e.g., removing the Bolon et al. 2022 and Phon-Amnuaisuk et al. 2016 citations). Also, a number of the additional citation would very much work to the benefit of this write-up (e.g. the Siers et al. 2024 (10.3897/neobiota.90.103041) and there are a number more across their review too.

Additionally, Reviewer 2 also brings up some good points with regards to maybe shifting some the snake categories to be more reflective of natural history (e.g., reclassifying brown watersnake) and also tightening up some of the estimates presented for the cost-benefit analysis. They even go as far to supply some R code that may help visualise some of your data (which for a reviewer really is above and beyond). Also, it appears that Table 1 may be from the first version, rather than the revised one. Just double-check that.

All in all, I think there are some really helpful suggestions and recommendations that Reviewer 2 is providing and I encourage you to give them all some thoughtful consideration as you revise this version.

Given the leaps and bounds this manuscript is taking, I very much look forward to reading the next version and feel that these recommended improvements will help you continue to polish this into an article that will be able to really have an impact.

I hope you find these reviews helpful and can apply the ideas being provided to continue to make this manuscript shine.

·

Basic reporting

The reporting in this new, more finely tuned MS is clearer, unambiguous and straightforward. This makes it a great improvement on the original MS.

Experimental design

The comments by the Editor and other reviewer on the non-standardized spatial and temporal factors of the experimental design have now been clearly explained and discussed.

Validity of the findings

This study gives clear conclusions, backed by a robust data set and statistically sound analyses.

Additional comments

Line 100: Python "molurus" has crept back in!
Line 298: is.. "other" species.. required here? I would delete "other".
Line 413-415: A clumsy sentence. Would a sentence like "Finally, evidence suggests that a combination of these cues is more efficient when using an inanimate prey lure" make more sense?

Fig 3, Caption: Add "of.... to "types snake"

·

Excellent Review

This review has been rated excellent by staff (in the top 15% of reviews)
EDITOR COMMENT
Reviewer 2 provided an exceptional review. It was detailed, thoughtful, and provided a lot of tremendous insight in how the author could improve their manuscript and ensure it was as impactful as possible. It is reviews like this that truly make the peer review system function as it should; providing comments and suggest that work to bolster the authors' writing and presentation of their work.

Basic reporting

I have identified several minor changes that are needed for clarity of language, and a handful of literature references that should be changed, omitted, or added, detailed in part 4

Experimental design

The experimental design is sound, although I have one major issue with the snake species categorization that should not affect the primary result. See part 4 below.

Validity of the findings

I have outlined several issues with the generality and interpretation of the findings in part 4 below

Additional comments

Thanks for your revisions. I have little doubt that this paper will be a useful contribution and should be published. However, some issues remain, which I have outlined below.

Title should be: Mammalian lures monitored with time-lapse cameras increase detection of pythons and other snakes (not “increases”)

Snake categorization

I don’t agree that brown watersnakes (Nerodia taxispilota) should be classified as a species with a high probability of being attracted to large mammalian lures. All pertinent literature indicates that N. taxispilota are piscivorous, eating almost exclusively fish (to a greater extent than other species of Nerodia) (see Table 17 on p. 205 of Gibbons & Dorcas 2004. North American Watersnakes for a summary of five studies with a total sample size >1700 N. taxispilota where fish dominated the diet and the only non-fish prey were a few frogs). The other species you designated as larger native snakes capable of eating mammals as large as rabbits all seem appropriate to me, although some (EDBs) are mammal specialists, most are generalists, and one (cottonmouth) is an ambush-forager that rarely feeds on mammals. In addition, diets of adult black racers include more mammals than those of cottonmouths (7-33% of racer diet in Kilmstra 1959, Halstead et al. 2008, v. 2-12% of cottonmouth diet in Himes 2003, Vincent 2004, and Burkett 1966), although even the largest black racers probably cannot eat adult rabbits. I would suggest that you omit watersnakes from this category and in doing so stick with the criterion is that the species should have the capacity to eat an adult rabbit. The nomenclature of your groups is also misleading, especially on lines 362-363 where “Like larger snakes and pythons, other snakes spent significantly more time at treatment pens” makes it sound like “other snakes” does not include the larger mammal-eating species (also true on lines 379 and 380-381). I would encourage the use of “pythons”, “mammal-eating native snakes” and “all native snakes” although you could also just drop the “all native snakes” part of the analysis because I don’t think it adds much. If anything adding Nerodia, Opheodrys, Thamnophis, Storeria and Diadophis weakens the trend (74% more likely when these snakes are included vs. 165% when they are excluded).

Generality of primary finding

My most important comment remains that these findings might only be generalizable to places where mammal population densities are low enough that preferred prey are a limiting/motivating factor. A very recent study clearly demonstrated that the attractiveness of mammalian baits to invasive snakes is significantly reduced in ecosystems where prey are not (yet) limiting, i.e. in the early stages of invasion. Siers et al. (2024. Limitations of invasive snake control tools in the context of a new invasion on an island with abundant prey. NeoBiota 90:1-33) found that traps baited with live and dead mouse lures were almost totally ineffective at catching Brown Treesnakes on Cocos Island, in contrast to on mainland Guam where they are among the most effective tools. We know that mesomammals have declined from historic baselines in ENP, although I’m not familiar with what data are available from C4 and Frog Pond. If you think it’s probable that there are very few wild mammalian prey available to the pythons and other snakes where you conducted the study, then I would argue that, taken together with the Siers paper, there is reason to believe that mammalian lures/baits could be more effective in your study than they would be in a newly-invaded area, with important implications for initial rapid response efforts in future invasions by snakes, which are likely to occur in prey-rich environments.

Other issues of interpretation

Your cost estimate for rabbit-lure camera surveys seems to double-count mileage ($17,565 to drive to sites on lines 371-372 and $4,593 for milage for travel to and from field sites on line 374). It’s also unclear to me why the visual surveys cost includes all rabbit related expenses. You should discuss your cost estimates in the context of other techniques ($298/python for contracted road cruising that mostly removes juveniles; $1,800-$11,000/python for scout snakes that target breeding females); see Table 4 of Guzy et al. (2023. NeoBiota 80:1-119).

Because you only set cameras May-August, you might want to qualify in the discussion that this finding applies only to the wet season when snakes are a bit more active in south Florida and it remains to be seen how effective live bait would be during the cooler dry season.

You may want to read and cite Gati et al. 2020. Assessment of Python Trapping within the Everglades Region Using a Patented Large Reptile Trap. Final Report to the Florida Fish and Wildlife Conservation Commission. Contract Number 13416-A3045. This is one of the only other python trapping studies there is so it seems like an omission not to cite it.

Line 106: Despite the general acceptance that pythons are ambush predators, there are few observations in the wild of this behavior. Whitney et al. (2021. Animal Biotelemetry 9:1-13) obtained continuous accelerometer data from four wild female Burmese pythons and found that pythons spent an average of 86.1% of their time resting. Resting periods were consistent with many ambush-foraging snakes (Pope 1961; Daniel 2002), but data are limited to female pythons during the breeding season, a time when females may move less than males. Further, pythons will actively search for prey, as evidenced by their consumption of wading bird nestlings and eggs (Orzechowski et al. 2019). Snake foraging strategies fall along a continuum of ambush to active strategies, and Burmese pythons engage in both strategies. In addition, I agree with Reviewer 1 that you should mention the possibility that the same python returned and was photographed night after night, which could limit the efficacy of this technique for trapping.

Lines 130-138: The statement “reduced detection rates are a function of Burmese pythons’ crypsis and use of aquatic environments” is definitely true in a general sense. However, you are contrasting the measurements of <5% from Nafus et al. 2020 with our estimates from South Carolina (3- 46%; Durso et al., 2011). There are a few problems with this contrast.
1. Estimates from Durso et al. 2011 are based on aquatic trapping so they are not directly comparable with visual encounter surveys.
2. Estimates from Durso et al. 2011 are of aquatic snakes, so the fact that pythons sometimes use aquatic habitats is not a valid reason why their detection probabilities would be lower than our estimates for other, even more aquatic species (Liodytes, Farancia). Pythons are less aquatic than most of the species we studied in the 2011 paper.
3. The estimates in Nafus et al. 2020 that are most comparable to those in Durso et al. 2011 are those in Table 3, where point estimates of detection range from 1.46% to 0.01% and upper 95% confidence estimates range from 2.78% to 0.5%. The estimate of 5% reported in their abstract comes from their Fig. 4 which shows that the upper confidence interval can reach ~5% at midday. It would be more accurate to report and compare the point estimates from Table 3.
4. Check Fig. 1 of Willson et al. (2023. Overcoming low detectability in snake conservation research: case studies from the southeast USA. Pages 61-76 in Walls and O'Donnell (eds.) Strategies for Conservation Success in Herpetology. Society for the Study of Amphibians and Reptiles, University Heights, Ohio) (I have provided a copy as the book is still in print) for other visual encounter survey studies of snakes and their associated measures of detection probability (corrected for effort). Nafus et al. 2020 is included and you can see that although the point estimate in Nafus is the lowest non-zero estimates for any snake, this estimate is not really an outlier. We took pains to extract comparable estimates and Melia Nafus converted their team’s effort measurements from person-km to person-hours for me so that we could include and fairly compare their values.
5. Are snakes that use aquatic or subterranean microhabitats less detectable than those that do not? I think the answer is yes, but to my knowledge no formal analysis has been conducted.
6. Are Burmese pythons more cryptic than other snakes? Maybe for their size, but in general I don’t think so. See also comments on ambush foraging above.
To be clear, I do think that Burmese pythons probably have lower detection rates than, say, black racers, but I’m not convinced their detection rates are lower than many other semi-aquatic or semi-fossorial snake species.

Line 455: While I appreciate you citing our paper, the model in Bolon et al. 2022 does not automatically detect photos with animals by estimating the change in pixels between time-lapse photos; rather, it can identify which species of snake is present in an image (but does not know how to differentiate images with snakes in them from those without). Instead I would direct you to the following:
Binta Islam, S., D. Valles, T. J. Hibbitts, W. A. Ryberg, D. K. Walkup, and M. R. Forstner. 2023. Animal Species Recognition with Deep Convolutional Neural Networks from Ecological Camera Trap Images. Animals 13:1526.
Walkup, D. K., W. A. Ryberg, J. B. Pierce, E. Smith, J. Childress, F. East, B. L. Pierce, P. Brown, C. M. Fielder, and T. J. Hibbitts. 2022. Testing the detection of large, secretive snakes using camera traps. Wildlife Society Bulletin:e1408.

In addition, one of the citations you give on lines 530-531 is not appropriate. Phon-Amnuaisuk et al. (2016. Computational Intelligence in Information Systems: Proceedings of the Computational Intelligence in Information Systems Conference) is a conference proceedings that contains a paper (Amir et al. 2016. Image Classification for Snake Species Using Machine Learning Techniques. Pages 52-59) about an algorithm that can identify a few species of snakes at the species level, not at the individual level. You should stick with just the Yang et al. 2013 citation here.

Finally, I think you could make more of an effort to review the unknown colubrid and unknown snake images, as they make up a large proportion (>50%) of your dataset. It’s difficult for me to say how realistic it is to identify more of them without seeing them and I’m sure the professional herpetologists you consulted are more than capable, but I can’t help but want to take a crack at it. If you’d be willing to share, I’d love to challenge my herpetology students with some of these impossible-to-identify images.

Minor comments:

Line 84: “often consider how…”
Line 93: “roof of the mouth”
Lines 106-108: Revise so that it reads “Live rodents have also been tested as bait for Burmese pythons (Python bivittatus) which aggregated near traps rather than entering them…”. Also, although it’s a good paper, I didn’t see what function the Emer et al. 2022 reference, which is about the physiology of python thermoreceptors, has in this sentence.
Line 109: omit comma after “mammalian lures”
Line 130: “an efficient and effective way”
Line 142: “different types of snakes” (not “differ”)
Line 191: although there are no official common names for reptiles, I haven’t seen “eastern corn snake” used for P. guttatus; I would recommend simply “corn snake”
Line 193: replace two spaces with one before “peninsula”
Line 199: braminus is now in the genus Indotyphlops, not Ramphotyphlops
Line 201: replace comma with colon after “categories”
Line 217: space between “Florida” and “USA”
Line 231-232: Add a citation for the scale (2 = thin)
Line 250: “independently each” is redundant, omit one
Line 305: “Whereas…” is an incomplete sentence, combine with the previous sentence
Line 322: I suggest “by dividing the total cost by the number of snakes…”
Line 326: replace “into” with “in”
Line 356-359: rephrase like “Time-lapse cameras at pens with rabbits were also 165% (IRR = 2.65, 95% CI 1.52- 4.62) more likely to detect larger snakes (β = 0.973, z-value = 3.42, p < 0.001), and these snakes spent more time at treatment pens (W = 250, p < 0.036), with a median of 8 mins compared to a median of 2 mins at control pens.”
Line 361: insert “all” before “snakes other than pythons”
Line 376: insert comma in $2835 for consistency
Line 406: remove comma after “additional”
Line 410: The phrase “some limited success” is pretty vague. Suggest citing Romagosa CM, Steury T, Miller MA, Guyer C, Rogers B, Angle TC, Gillette RL (2011) Assessment of detection dogs as a potential tool for python detection efforts. Final report submitted to the National Park service-Everglades National Park (Contract H500002A271) and South Florida Water Management District (Contract 4500059338).
Line 418: “daylight” should be one word
Lines 439-441: This seems reasonable to me
Line 478: I would suggest replacing “may” with “could” – I’m skeptical that a prey replica without scent would be likely to attract snakes

Figures and tables

The changes to the scientific names in Table 1 indicated on page 15 of the rebuttal have not been made (A. piscivorus -> A. conanti, P. molurus bivittatus -> P. bivittatus, L. floridana -> L. getula, combine Nerodia fasciata fasciata and Nerodia fasciata confluens). It might also make sense to include a column in this table indicating which species are included in the “large mammal-eating” category in Fig. 3.

In Fig. 2, it might be useful to show the cage up close in a sub-panel

In Fig. 3, it would be more informative to label the black bars “baited” and the gray bars “unbaited”, or at least “baited (treatment)” vs. “unbaited (control)”. This is a tiny change but I think this is worthwhile because many readers will just look at the figures so it makes it clear what your control and treatment are. I thought the rabbit icon you had before was good, just in the wrong place.

Here’s my suggestion to show the visual survey period on Fig. 4, using your code as a basis:

ggplot(dat) +
geom_histogram(aes(x = hours),
bins = 24,
boundary = 0,
color = "white") +
scale_x_continuous(name = "Time (hours)",
breaks = c(seq(0, 24, 4)),
labels = paste0(c(seq(0, 24, 4)), ":00")) +
ylab("Pictures") +
theme_bw() +
theme(panel.grid = element_blank(),
axis.text = element_text(size = 18),
axis.title = element_text(size = 24)) +
geom_vline(xintercept = 7, linetype = 2) +
geom_vline(xintercept = 15, linetype = 2) +
labs(x='Time of day (hours)', y = 'Number of photos')+
geom_curve(aes(x = 7, y = 230,
xend = 9, yend = 250), curvature = -0.2, colour = "grey25") +
geom_curve(aes(x = 13, y = 250,
xend = 15, yend = 230), curvature = -0.2, colour = "grey25") +
annotate("richtext", x = 11, y = 250, hjust = 0.5, lineheight = 0.85,
colour = "grey25",
label = "Timespan of
visual sampling",
label.padding = grid::unit(rep(0, 4), "pt"),
fill = NA, label.color = NA)+
geom_vline(xintercept = 18, linetype = 3) +
geom_vline(xintercept = 20.25, linetype = 3) +
geom_curve(aes(x = 18, y = 230,
xend = 18.5, yend = 250), curvature = -0.2, colour = "grey25") +
geom_curve(aes(x = 19.75, y = 250,
xend = 20.25, yend = 230), curvature = -0.2, colour = "grey25") +
annotate("richtext", x = 19.125, y = 250, hjust = 0.5, lineheight = 0.85, size = 3,
colour = "grey25",
label = "Timespan of
canine
sampling",
label.padding = grid::unit(rep(0, 4), "pt"),
fill = NA, label.color = NA)

I think this is well worth it because many readers will not read the entire article, they will just look at the figures. I would also suggest reminding readers in the caption to Fig. 4 that these data are for May-August.

Fig. 4 of Nafus et al. 2020 also shows an inverse pattern of diel activity from December to April, it would be worth mentioning this along with the language on lines 421-423 because someone trying to replicate or apply your results would want to know about the seasonal change from nocturnal to diurnal activity.

Data and code:

Error when running lines 73 and 76 of code:
Others_glmm <-glmmTMB(S ~ Treatment + (1|Pair), family=poisson, data=snakes3)
Error in fitTMB(TMBStruc) : negative log-likelihood is NaN at starting parameter values

On line 95, dat$hours = as.numeric(dat$Time)/(60*60) but column is called timestamp, not Time

---

## Round 0.3 · Minor Revisions

I think this manuscript is looking good and that the review process and the effort put in by the authors and reviewers have really help refine and polish this article. There we a few aspects however that were left out after the last round of revisions that both myself and Reviewer 2 have noted. In reality, I think all that is really needs is the addition of some text to ensure that recent and/or relevant information that can give some deep context to this study is included.

I recognise that during the last review the authors noted that they felt several of these suggestions were not applicable, but I would strongly suggest the authors reconsider their stance. For example, invasive brown tree snake management has had a tremendous amount of research effort put into it over the years and not including findings from papers on it because the authors’ feel it is “a very different snake, a very different system, and a very different prey base” is not really a valid reason to exclude some discussion on their findings in relations to what this work found. Yes, there will be differences, but those are worth discussing, just as there is absolutely relevant information that helps to give your findings better context. Similarly, Reviewer 2 also lists several other studies that looks at how baiting increases capture rates in snakes. As such, I think it would do this manuscript a service to including some text discussing the findings from:

1) Siers et al. 2024. Limitations of invasive snake control tools in the context of a new invasion on an island with abundant prey. NeoBiota 90:1-33 (this is worth devoting a few lines of text to for sure)

2) Keck 1994. Herpetological Natural History 2:101-103;

3) Winne 2005. Herpetological Review 36:411-413

4) Rodda & Fritts 1992. Micronesica 25:23-40

5) Rodda et al. 1999. A state-of-the-art trap for the Brown Treesnake. Pages 268-305 in Rodda et al., editors. Problem Snake Management: The Habu and the Brown Treesnake. Cornell University Press, Ithaca, NY

Furthermore, given that other research has found a relationship between prey availability (as a product of density) and trapping success, it is worth discussing this. The purpose of such discussion in your manuscript is because your future readers may well have many of the exact same sort of questions being raised by your reviewers. Our goal is to anticipate those questions, and have answer for them ready in the manuscript. So, given future readers may have questions about prey availability (given this python has been associated with mammal declines in the Everglades) and given that reader may also be familiar with the idea of prey densities impacting trap success, I once again agree with Reviewer 2. I suggest that you include information stating that wild mammals are still present at these two sites and provide any specific information you can (if nothing is known about whether their densities resemble pre-python densities this should be stated). Again, the idea is, if it raises questions in the reviewers, it very much will raise questions in our readers. Thus, we should aim to provide additional info in the manuscript to address those question ahead of time (i.e., a nip it in the bud approach). That way your reader will know you considered the same sort of factor they are wondering about and can read your reasons/justification for how it does or does not relate/impact your findings – essentially the more context the better.

I ask the authors to thoughtfully read through Reveiwer 2’s comments and suggestions and work to address them directly in the manuscript. For the addition of a line of text, you can make the case for why some of the BTS data works and given context, and why some of it does. Similarly, why prey density may or may not be an issue. And so on. But looks to what the Reviewer flag, view it is a potential area of ambiguity or uncertainty for your future reader, and then add text to make sure that ambiguity or future question is address in the text.

I do not imagine this should take too much time and I am very much looking forward to reading the next draft and feel that with just a little more work this manuscript will be ready for acceptance shortly.

·

Basic reporting

I have suggested a few more important literature references, please see the attached PDF.

Experimental design

No comment

Validity of the findings

My most important comment is that the findings must at a minimum be placed into context of what is known about current vs. historic mammal density at the study sites, and if nothing is known, then that should be stated. I have included more detailed responses to the authors in the attached PDF in green.

Additional comments

My comments are included in the attached PDF in green.

---

## Round 0.4 · Minor Revisions

The authors have done a great job and have made all of the requested edits. The manuscript is looking great. During my last read I caught a handful of typos, however, and just to make sure they do not travel through copy editing and proofing, I thought I'd flag them now. So if you could just go through and accept the edits then I can proceed to recommending we accept it. Most are super small (e.g., double spacing or no spacing or a semicolon left hanging or a forgotten oxford comma here and there), also "Python molurus bivittatus" snuck in again instead of "Python bivittatus". Additionally, the link to the figshare repository needs to be adding in for real (rather than the placeholder).

I will email the corresponding author a tracked changes word file directly to expedite the editing process and we can have this wrapped up by the end of the week.

---

## Round 0.5 · accepted · Accept

The manuscript is looking good and has definitely been polished and leveled-up during the review process. I hope the authors are feeling pleased with the finished product.